# Comparative Analysis for the Distinction of Chromophobe Renal Cell Carcinoma from Renal Oncocytoma in Computed Tomography Imaging Using Machine Learning Radiomics Analysis

**DOI:** 10.3390/cancers14153609

**Published:** 2022-07-25

**Authors:** Abeer J. Alhussaini, J. Douglas Steele, Ghulam Nabi

**Affiliations:** 1Division of Imaging Sciences and Technology, School of Medicine, Ninewells Hospital, University of Dundee, Dundee DD1 9SY, UK; d.steele@dundee.ac.uk; 2Department of Medical Imaging, Al-Amiri Hospital, Ministry of Health, Sulaibikhat 1300, Kuwait

**Keywords:** chromophobe, computed tomography, machine learning, oncocytoma, radiomics, renal masses

## Abstract

**Simple Summary:**

Chromophobe renal cell carcinoma (ChRCC) is the 3rd most common subtype of renal cell carcinoma (RCC), which is difficult to differentiate from benign renal oncocytoma (RO) using conventional imaging techniques. The aim of this study was to differentiate chromophobe renal cell carcinoma and renal oncocytoma non-invasively using radiomics features extracted from pre-operative computed tomography (CT) scan images in combination with machine learning (ML) techniques for classification. This would help in providing “virtual biopsy” and may prevent unnecessary surgical resection for benign renal oncocytoma tumors.

**Abstract:**

*Background:* ChRCC and RO are two types of rarely occurring renal tumors that are difficult to distinguish from one another based on morphological features alone. They differ in prognosis, with ChRCC capable of progressing and metastasizing, but RO is benign. This means discrimination of the two tumors is of crucial importance. *Objectives:* The purpose of this research was to develop and comprehensively evaluate predictive models that can discriminate between ChRCC and RO tumors using Computed Tomography (CT) scans and ML-Radiomics texture analysis methods. *Methods:* Data were obtained from 78 pathologically confirmed renal masses, scanned at two institutions. Data from the two institutions were combined to form a third set resulting in three data cohorts, i.e., cohort 1, 2 and combined. Contrast-enhanced scans were used and the axial cross-sectional slices of each tumor were extracted from the 3D data using a semi-automatic segmentation technique for both 2D and 3D scans. Radiomics features were extracted before and after applying filters and the dimensions of the radiomic features reduced using the least absolute shrinkage and selection operator (LASSO) method. Synthetic minority oversampling technique (SMOTE) was applied to avoid class imbalance. Five ML algorithms were used to train models for predictive classification and evaluated using 5-fold cross-validation. *Results:* The number of selected features with good model performance was 20, 40 and 6 for cohorts 1, 2 and combined, respectively. The best model performance in cohorts 1, 2 and combined had an excellent Area Under the Curve (AUC) of 1.00 ± 0.000, 1.00 ± 0.000 and 0.87 ± 0.073, respectively. Conclusions: ML-based radiomics signatures are potentially useful for distinguishing ChRCC and RO tumors, with a reliable level of performance for both 2D and 3D scanning.

## 1. Introduction

Renal Cell Carcinoma (RCC) has the highest mortality rates of all genitourinary malignancies, and its prevalence has been gradually increasing [1,2,3]. In recent years, there has been an upsurge in the number of cases of RCC all over the world [4,5,6,7,8]. RCC incidences have continuously increased by 2–4% per year, and RCC is now the seventh most common cancer type in the United States [9]. According to Sung et al. [10], kidney cancer was mainly accountable for 431,288 clinically diagnosed cases and 179,368 deaths in 2020 globally. In the UK alone, kidney cancers accounted for 4% of all cancer occurrences between 2016 and 2018. Twelve deaths were reported daily in 2017 in the UK [11] and RCC contributed immensely to these statistics, as it is responsible for more than 90% of kidney cancers [12].

Renal tumors are quite diverse, having at least 16 distinct subtypes [13], out of which chromophobe renal cell carcinoma (ChRCC) and renal oncocytomas (RO) are very similar to each other. ChRCC is responsible for at least 5% of the diagnosed malignant renal tumors each year [14,15]. In contrast, RO accounts for 3–7% of all benign renal tumor diagnoses [16]. RO was initially characterized by Zippel in 1942 [17,18], whereas ChRCCs were first described by Theones et al. in 1985 [19,20]. Because ChRCCs were described four decades later than ROs, many renal tumors that were suspected to be ChRCCs were characterized as ROs throughout that period [13]. Nestled pattern, myxoid stroma, granular cytoplasm, and round nuclei are all likely signs of RO, whereas varied nuclear size, raisinoid nuclei, and reticular cytoplasm are more likely signs of ChRCC. Typically, RO cells have round nuclei, but in an investigation of RO cells, a raisinoid nuclei was observed, which is a key feature of ChRCCs [21]. Surprisingly, components of RCC can be seen in 10–30% of ROs; hence, the presence of an RO in a sample does not confirm the absence of renal cancer [21]. So, there is a clinical challenge in identifying RO from ChRCCs in a given sample.

Various conventional methods have been used to diagnose and differentiate between these two highly similar subtypes of renal masses such as biopsy, MRI and CT scans. Each of these methods has limitations in diagnosing and differentiating the two. Currently, no proposed CT scan markers can reliably distinguish ROs from RCCs. As a result, most ROs are classified as suspicious of RCCs on the basis of imaging and are usually exposed to surgical excision [22]. Similarly, research on the potential of MRI to identify ROs from ChRCCs concluded that both groups had comparable characteristics, and no MRI clinical features could help differentiate the two [23].

Unlike MRI and CT scans, a renal mass biopsy presents an opportunity for a pre-operative diagnosis. However, this approach has various potential problems, making surgical resection unavoidable. One of the significant disadvantages of using biopsy is that it is difficult for a pathologist to diagnose renal tumor subtypes accurately from insufficient tissue biopsy samples, as a whole range of cyto-architectural features are usually required for analysis to come to a diagnosis [24]. Generally, a lesion is reported as ChRCC if it looks exactly the same as a chromophobe in the needle biopsy. However, if the pathologist identifies the lesion as RO in the needle biopsy, it is concluded that more tissue sampling for diagnosis is needed because of tumor heterogeneity. This is because there are many variants of ChRCC that are more similar to RO than to ChRCC. In addition to the difficulty in clinically distinguishing ROs from ChRCCs, the characteristics of these pathological tumors following renal biopsy sometimes coincide, making diagnosis particularly problematic for pathologists [13].

Moreover, the available methods of tumor identification are not conclusive, as they are subjective. Likewise, a biopsy, which is the method in use currently, although accurate, is an invasive technique that has its own limitations [25]. On the other hand, according to the literature [26,27,28], the prevalence of benign tumors ranges between 13 to 30% of all surgically resected lesions as the possibility of benign renal histopathology in small renal masses is determined by the size, with about 40% of the tumors being smaller than a centimeter in diameter [27]. This, as a result, further leads patients to undergo expensive and unnecessary surgery.

Recent studies show that ROs and ChRCCs have similar histological and cytologic characteristics and immunohistochemistry (IHC) markers for S100A1 and CD117 KIT [29]. However, varied forms of renal tumors act differently and have different prognoses. They may be difficult to distinguish due to some overlapping morphological traits and immunohistochemical staining patterns [29]. Similarly, non-diagnostic core-needle biopsy and errors due to sampling, both quite typical with percutaneous biopsy, are limiting factors in correctly diagnosing these two RCC subtypes [30]. Therefore, it is essential to tell the difference between ChRCCs and ROs before surgery to manage a patient’s condition better.

Due to the challenges in differentiating RO from ChRCC clinically and histopathologically through biopsy [29], there is a need to develop a more accurate, reliable, and clinically applicable method in differentiating RO from ChRCCs. Recently, there has been technological advancement in medical imaging, enabling medical researchers to capture tissue anatomy characteristics, physiological functions and quantitative features through images that help in precision medicine [31]. The advantage of this is that non-invasive methods of tumor identification have been investigated, hence assisting in solving the shortcomings of biopsy and efficiently detecting tumor differences.

Quantitative imaging is now possible through advancements such as improved technology, imaging agents, and standardized protocols. Radiomics is the recent variety of medical imaging signature breakthroughs, focusing on image analysis enhancements, employing automatic high-dimension extraction of large volumes of quantitative aspects of medical image data [32,33]. Ten years ago, Lambin et al. [32] proposed the possibility of extracting radiomics features based on the differences in solid renal tumors. By extracting such features from high-dimensional image data, valuable meaningful information can be extracted instead of visually observing the features [32]. Many studies have investigated the potential of radiomics texture analysis as an alternative to the traditional imaging methods of differentiating RO from ChRCC. However, these studies have focused on the theoretical aspects rather than the practical application of radiomics texture analysis. Likewise, there is limited research on the use of radiomic feature analysis on rare types of renal tumors. Moreover, according to our knowledge, no paper has attempted to investigate the effect of filter features as well as a hybrid study, i.e., the combination of both prospective and retrospective research in a single study on the accuracy of radiomic models. The use of all tumor slices for the prediction of patient histopathology has also not been investigated before. This is the first paper, according to our knowledge, that has used the highest number of participants in the differentiation of ChRCC from RO using ML-based radiomic signature.

Therefore, in this research, the prospects of a hybrid study, effects of filter features and all tumor slices analysis combined with ML techniques have been investigated in an effort to differentiate RO from ChRCC in order to develop better non-invasive pre-operative diagnostic models than the traditional methods.

## 2. Materials and Methods

### 2.1. Ethical Approval

The East of Scotland Research Ethical Service approved this study. Patients’ medical health care data were accessed under the Caldicott Approval Number: IGTCAL9519.

### 2.2. Patients

The research conducted a multi-center study of two institutions. The first cohort was a prospective protocol-driven analysis from an actively maintained database of 35 Patients (10 ChRCC and 25 RO) from Ninewells Hospital between 2011 and 2021. In the second cohort, a retrospective analysis was conducted on 43 patients (27 ChRCC and 16 RO) from The University of Minnesota [34,35] (Kits-Challenge 2019) between 2010 and 2018, leading to a total sample size of 78 (37 ChRCC and 41 RO). The cases from both cohorts had been pathologically confirmed from their respective institutions. The participants in the study had received pre-operative contrast-enhanced CT scan imaging. However, the first cohort used the nephrographic phase, whereas the second cohort used the arterial phase. In the first cohort, the age range was between 43–89 years with an average age of 66 years. Gender wise, there were 24 males and 11 females. The data was provided through the institutions’ Picture Archiving and Communication System (PACS) in DICOM format of size 512 × 512 × 3 pixels. In the second cohort, the ages ranged between 21–83 years with an average age of 52 years. There were 14 males and 29 females. The data was obtained through their public repository [34] in NIFTI format.

### 2.3. CT Scan

Images from cohort 1 were captured using a Helical CT scanner (GE Healthcare). The parameters included a large body scan field of view (SFOV), 0.7 sec gantry rotation time, 1.25 mm slice thickness, 1375:1 pitch, 40 mm detector coverage, Noise Index (NI) 30, Computed Tomography Dose Index Volume (CTDIvol) 9.59 mGy, X-ray tube-voltage 120 kVp, X-ray tube-current 100–560 mA (auto-modulated) depending on the patients’ size. The contrast agent used is intravenous Omnipaque 300 (80–100 mls as standard per patient). The contrast pressure injector used is by the manufacturer Bayer and the model is Centargo. The flow rate of contrast injection for the renal scan is 3 mL/s. The crucial pre-operative CT nephrographic stage [36,37,38,39,40,41,42,43] that arises 100 to 120 s after IV contrast injection was used for this study; it provided the clearest identification of renal lesions. There was no standard protocol for image capturing in cohort 2 [34,35].

### 2.4. Segmentation

De-identified DICOM image slices from cohort 1 were converted to 2D JPG format resulting in 967 slices. In addition, the DICOM image slices were also converted to 3D NIFTI format for each patient using Python software for 3D tumor segmentation. Segmentation of the 2D slices was done manually by contracting the edges of the tumor by about 2 mm and delineating the region of interest (ROI). This was done after image grey level conversion and a Wiener filter with kernel size 2 × 2 had been applied using MATLAB software version 9.10. A Wiener filter is a filtering technique used to reduce noise and for image reconstruction to improve medical image quality. Khudayer Jadwa et al. [44] proposed a Wiener filter-based noise reduction method as an effective approach to enhance the image quality from CT and Magnetic Resonance Imaging (MRI) [45]. The 2D segmentation was carried out as shown in Figure 1.

Finally, 3D volume of interest (VOI) segmentation was done semi-automatically using the Slicer 3D software version 4.11, where the VOI was automatically generated after segmenting the first few slices. The 3D segmentation was done as shown in Figure 2. Both 2D and 3D segmentations were performed by a blinded experienced investigator (A.A) delineating the ROI about the edges of the tumors without prior knowledge of the patient’s condition. After that, the segmented masses were revised and confirmed by an experienced urological surgical oncologist (G.N), who further took into consideration the notes of histology and radiology reports. The study used the histopathological assessment after biopsy or nephrectomy as the gold standard.

For cohort 2, manual delineation of renal tumor boundary was done by students under the supervision of a clinician using a web application in an HTML5 canvas developed in-house by the University of Minnesota. Refinement and the final check of delineated boundary was done by Nicholas Heller taking into consideration radiology and pathology notes [46]. Our research then extracted 1431 2D JPG slices from their resulting 3D segmentation using the Python programming language version 3.9.

### 2.5. Different Strategies Applied:

The cohort 1 and 2 data were put together to create a new cohort called the Combined Cohort. Based on the segmentations, different scenarios were implemented in order to assist in distinguishing ChRCC from RO in both 2D and 3D imaging. First, the mid cross-sectional CT slice was extracted from each tumor volume in each of the cohorts to form the *Largest Tumor Slice* category. Secondly, all the tumor axial slices in each cohort formed a category called *All Tumor Slices*. The analysis of all tumor slices further gave rise to another category called *Per Patient Prediction*. The 3D scans in each cohort also formed a different category called *Whole Tumor Volume*. For each of these categories of data, original and filtered features were to be extracted, as will be explained in the proceeding sections. The diagrammatic visualization of the process is indicated in Figure 3.

### 2.6. Feature Extraction

After segmentation of the ROI from 2D JPG slices, quantitative data features based on 6 classes of radiomic features were extracted automatically using the PyRadiomics library [47] available within the Python foundation software version 3.6.1. To better understand the representation of the tumor; quantitative features were extracted from all the slices for each patient using a fixed bin-width of 25. The classes of features were as follows: The first-order statistic, Gray-Level Co-occurrence Matrix (GLCM), Gray-Level Run-Length Matrix (GLRLM), Gray-Level Size-Zone Matrix (GLSZM), Neighbouring Gray-Tone Difference Matrix (NGTDM) and Gray-Level Dependence Matrix (GLDM). These feature classes explain the spread of texture intensity in the CT slices. Refer to Table 1 for the description of feature classes. Currently, no standard protocol exists on the kind of radiomics features to be extracted for feature analysis, and this affects the repeatability and consistency of the results. PyRadiomics, therefore, has attempted to come up with a standardized process for medical images feature extraction [47].

Radiomic features were also extracted from the 3D VOI. In addition to the 6 classes of features extracted for 2D, the shape class feature was also extracted to capture the three-dimensional feature of the delineated regions.

Radiomic feature extraction using the PyRadiomics library was in two folds. First, the study extracted original features without applying any filters and secondly, extracted features after eight classes of filters had been applied. These filter classes included: Wavelet filter, Laplacian of Gaussian filters (LoG), Square, Logarithm, Square-Root, Gradient Exponential and Local Binary Pattern 2D/3D (LBP). Refer to Table 2 for the description of filter classes.

### 2.7. Feature Pre-Processing and Selection

The data was analyzed for null values and in-cases where nulls were found the sample was removed from the data. The radiomics features were normalized using a standard scaler so that the mean of each feature is zero with a standard deviation of one. The ground truth labels were annotated as 1 and 0 for RO (negative class) and ChRCC (positive class), respectively, in preparation for classification. The least absolute shrinkage and selection operator (LASSO) model [48,49,50,51] was used in selecting the essential features. Alpha (α) parameter value determined the coefficient of features, the higher the (α) value, the lower the coefficient of features and vice versa. The coefficient for each feature was calculated using 5-fold cross-validation. Features with coefficients greater than zero were retained for modeling. The cost function of LASSO was calculated as in Equation (Equation 1) for feature coefficients [52].
(1)$$Cost\left(w\right)=\sum_{n=i}^{\infty }{\left(y_{i}-\sum_{n=i,j}^{\infty }X_{i,j}\beta_{j}\right)}^{2}+\alpha \sum_{n=i}^{\infty }\left|\beta_{j}\right|$$
where,

α denotes the amount of shrinkage;$y_{i}$ denotes the true label;$X_{ij}$ denotes the features;$B_{j}$ denotes the slope of variable j;*n* is the sample size.

### 2.8. Subsampling

The study data was highly imbalanced for both cohort 1 and 2 with the percentage ratio of ChRCC: RO being 29:71 and 63:37, respectively. To mitigate the challenges of imbalanced datasets in the study, synthetic minority over-sampling technique (SMOTE) [53] was applied to generate synthetic data, which reflects the structure of the original data from the minority group.

### 2.9. Statistical Test

A statistical test on the data was conducted using the SciPy package in Python software. Comparison between age, tumor size with maximal axial dimension, gender and histopathology were investigated to determine any significance. Tests assumed a significance level of 0.05. The Chi-square test and the Student T-test were used to assess the difference between groups. The confidence interval (CI) of accuracy, sensitivity, specificity and AUC were also computed using Z-test at 95% CI. The radiomic quality score (RQS) [54,55,56] was also calculated to evaluate whether the research followed the scientific guidelines of radiomic studies. This study followed the guidelines of transparent reporting of a multi-variable prediction model for individual prognosis or diagnosis (TRIPOD) which is available at https://www.tripod-statement.org (accessed on 8 April 2022) [55,57].

### 2.10. Model Training and Evaluation

The study used 3 cohorts, 2 feature types, 3 categories and 5 ML classifiers, namely: RF with 200 trees, support vector machine (SVM) with a linear kernel, K-nearest neighbour (KNN), logistic regression (LR) and naive bayes (NB), resulting in 3 × 2 × 3 × 5 = 90 distinct algorithms. Five-fold cross-validation was used for testing the efficiency of models due to the low sample size. The evaluation metric used included accuracy, sensitivity, specificity, and area under the receiver operating characteristic curve (AUC/ROC). Refer to Figure 4 for the diagrammatic representation of the methodology.

The Per Patient Prediction analysis of all tumor slices was treated as a special case as no machine learning algorithm was trained on it directly, but majority voting was done on the All Tumor Slices predictions for each patient to determine the final classification. Therefore, the final outcome of a patient was the tumor subtype in which most of the patient’s slices were predicted by each respective model. In cases where majority voting was not able to accurately distinguish the classes, i.e., when both classes have the same number of predictions, it is concluded that the patient tumor subtype is wrongly predicted.

## 3. Results

### 3.1. Statistical Analysis

Statistical analysis was conducted on patients’ age, tumor size and gender. From the analysis, it was found that there was no significant difference between tumor size (p=0.89), gender (p=1) or age (p=0.22) and histopathology in cohort 1 while, in cohort 2 and combined cohort, age and tumor size were significant. The study conducted a Pearson correlation coefficient test to determine whether the two significant variables have any effect on the model performance. However, the correlation between the prediction of the best performing model with age and tumor size was found to be insignificant and, hence, did not affect model performance. Detailed results are shown in Table 3. The RQS for the entire data set was found to be 69.7%, signifying that the research followed scientific radiomic guidelines. The RQS rubric used can be found at https://www.radiomics.world/rqs2 (accessed on 27 April 2022) [54,55].

#### 3.1.1. Feature Pre-Processing, Extraction and Selection

Null values were found in cohort 2 data set reducing its sample size to 42 in all categories except for Whole Tumor Volume. There were 95 and 109 features extracted from the 2D and 3D original data, respectively. Likewise, for filtered radiomic features, the total features extracted were 1304 and 1876 for 2D and 3D data, respectively. SMOTE was used to increase the number of samples from 967 to 1312, 1431 to 2152 and 2398 to 2774 for All Tumor Slices for cohorts 1, 2, and combined, respectively. LASSO model was used to perform feature reduction on each of the data sets. After feature reduction, the total number of features was reduced as listed in Table 4, Table 5 and Table 6. These features showed the highest ability to discriminate between ChRCC and RO. The process by which LASSO estimated the best (α) parameter and selected the best features for both 2D and 3D (original and filtered) for cohort 1 is displayed in Figure 5 and Figure 6, respectively.

#### 3.1.2. ML Model Diagnostic Performance

Several evaluation metrics were obtained from radiomics ML models using 5-fold cross-validation. Table 7, Table 8, Table 9 and Table 10 represent the different model performances in each category. A summary of the best diagnostic performance is displayed in Table 11. The best performance in cohort 1 was in the prediction per patient from the whole tumor slices original feature with an AUC value of 1.00 ± 0.000. For cohort 2, the highest AUC was in largest tumor slice with a value of 1.00 ± 0.000, whereas in the combined cohort, the whole tumor volume with filters exhibited the best performance with an AUC of 0.87 ± 0.073. Refer to Appendix A Figure A1, Figure A2, Figure A3 and Figure A4 for the best AUC plots from different cohorts and categories.

Largest Tumor Slice

All Tumor Slices

Per Patient Prediction from All Tumor Slices

Whole Tumor Volume

Best Model Performance

## 4. Discussion

ChRCC was formally categorized as a type of renal tumor in 1998 by the World Health Organization (WHO) [20]. ChRCC is the third most common type of RCC, which has a high tendency to metastasize. RO is the most frequent type of benign tumor and was discovered in 1942 [58]. RO accounts for between 3% to 7% of all diagnosed renal tumors. RO characteristics mimic those of RCC; hence in most cases, they are diagnosed incidentally, as they are mistaken for ChRCC [59]. Moch and Ohashi [20] assert that RO has a morphological heterogeneity similar to that of ChRCC. Baghdadi et al. [26] stated that both ChRCC and RO have CD117 (+) protein biomarkers, unavailable in other RCC tumors; hence, it is difficult to distinguish between the two tumors, as their morphological characteristics overlap.

Currently, the most effective treatment method for renal tumors is surgical resection. However, both radical and partial nephrectomy have complications. For instance, radical nephrectomy increases the chances of chronic renal diseases, which leads to cardiovascular diseases and mortality [16]. Research shows that up to 30% of surgically resected renal masses are ultimately benign [26], leading many patients to undergo unnecessary surgery. Biopsy is the most popular pre-operative examination technique, with almost 97% accuracy for differentiating malignant from benign renal masses in general [16]. However, renal biopsy has specifically reported difficulties in differentiating ChRCC from RO [20]. Therefore, this present research focused on ML-based radiomic analysis to distinguish between ChRCC and RO.

Radiomic analyses refers to the calculation of high dimension texture features using complex image processing technologies to obtain quantitative texture representations [32]. Using radiomics analysis, very important but small differences that are not detectable visually can be extracted and analyzed [31]. Radiomics, sometimes also referred to as a “virtual biopsy“, is advantageous in several ways, as it can capture both intra-tumoral (within-tumor) and inter-tumor (between-tumors) heterogeneity, and can be performed multiple times as opposed to biopsy. Therefore, this advanced image processing technique is potentially a more objective method for tumor analysis.

Experimental results from this study indicated that for cohort 1, the per patient prediction with original features had the best predictive performance with an AUC of 1.00 ± 0.000. In cohort 2, largest tumor slice with filter features had the best performance with an AUC of 1.00 ± 0.000. Finally, in cohort 3, whole tumor volume with filters exhibited the best performance with AUC of 0.87 ± 0.073. Filtered features were better than the original features for most models in their ability to distinguish ChRCC from RO. Overall, whole tumor volume was the best category to represent the heterogeneity of the tumor. SVM, KNN and RF models all offered promising results in radiomic feature analysis.

The combined cohort had the lowest diagnostic performance in all the categories. This is likely due to the fact that the cohort is a combination of two different data sets captured using different scanners and protocols. This led to poor generalization compared to the other cohorts. However, it is worth noting that the least performance was in the largest tumor slice with an AUC of 0.73, while the best performance was in whole tumor volume with AUC 0.87. Therefore, we concluded that in general the model generalized well in the multi-center study.

There exist limited studies on the differentiation of ChRCC from RO. This is largely because ChRCC and RO are rarely occurring renal tumors compared to other renal tumor subtypes. Therefore, most studies focus on the analysis of clear cell RCC (ccRCC) and papillary RCC (PRCC), which are more common. Sun et al. [60] implemented an SVM recursive feature elimination (SVM-RFE) classifier with 100 samples (64 malignant and 36 benign) to differentiate between malignant and benign renal tumor subtypes consisting of one group of PRCC and ChRCC versus another of angiomyolipoma without visible fat (AMLwvf) and RO. The model yielded a sensitivity of 83.3% and a specificity of 91.7%. In the paper by Sun et al. [60], 11 features were extracted from the CECT scan, which was then applied to the ML algorithm. Erdim et al. [61], just like Sun et al. [60] did not focus on a single malignant and benign renal cell subtype. However, the paper compared the performance of several ML radiomic feature analysis models between unenhanced and contrast-enhanced CT phases. The paper conducted a study using a total sample size of 84 renal tumors consisting of 63 malignant (ccRCC, PRCC and ChRCC) and 21 benign (AMLwvf and RO) [61]. These two studies are general in nature and cannot be used as diagnostic predictors in the differentiation of ChRCC from RO; as such, our study has limited the scope of tumor subtypes to only the rarely occurring tumors with similar morphological characteristics, i.e., ChRCC and RO.

Li et al. [16] explored how enhanced CT quantitative feature analysis can be used for the differentiation of RO from ChRCC. The paper’s authors conducted a retrospective study using 61 (17 RO, 44 ChRCC) pathologically confirmed cases of renal tumors. The paper implemented five ML algorithms for corticomidulary-phase (CMP), nephrographic-phase (NP), excretory-phase (EP) and combined CMP with NP, out of which the SVM classifier had the highest accuracy of 0.945. This was done after applying the LASSO technique for feature selection [16]. Whereas our paper focused on the differentiation of RO from ChRCC, the central point of departure from the paper by Li et al. [16] was that our research was based on the comparison between radiomic analysis for original and filtered radiomic features. Nonetheless, our study went further and did a 2D maximal axial tumor slice radiomic feature analysis in addition to the 3D analysis. All tumor slices were also analyzed and a majority voting technique was used to perform the per patient prediction. Through such analysis, we were able to determine the best criteria for radiomic feature analysis. Feng et al. [62] indicated that the use of a small data set, especially due to class imbalance, increased over-fitting and recommended using SMOTE to mitigate such challenges. Li et al. [16] did not describe how they mitigated the class imbalance in the data; this was addressed in our paper by implementing SMOTE. Moreover, the paper never looked at the prospect of a prospective and multi-center study as an alternative and even a better discriminant compared to a retrospective and single center study. In our research, this was adequately tackled by conducting both retrospective and prospective research as well as a single center and multi-center study.

Baghdadi et al. [26] investigated the possibility of using Artificial Intelligence (AI) in combination with an image processing signature in a semi-automatic design, using the tumor to cortex peak early phase enhanced ratio (PEER) to distinguish ChRCC from RO using convolutional neural network (CNN) segmentation in CT images. The authors had 192 participants for the training cohort and 20 for the testing cohort. As opposed to Baghdadi et al. [26], our paper analyzed both 2D and 3D images. Baghdadi et al. [26] did not investigate the possible importance of radiomics texture analysis for the purpose of differentiating renal tumors.

Uchida et al. [63] developed a diffusion coefficient map to assist in the distinction of RO from ChRCC using MRI texture features. The research focused on the analysis of important texture features in 3D MRI volume. The sample size of the study was 49 (ChRCC:41, RO:8); despite the small sample size and class imbalance in the data, the authors did not attempt to mitigate or solve the problem, this could have possibly affected the model performance leading to over-fitting.

Li et al. [16] suggested the use of contrast-enhanced CT images to increase the accuracy of classification models. Kocak, Ates et al. [64] analyzed the importance of edge segmentation on the performance of a model. The authors concluded that contracting the tumor edges of segmentation by about 2mm leads to better reproducibility and model performance [64]. In our paper, both manual and semi-automatic methods were explored for the purpose of segmentation. Lee et al. [65] developed a RF algorithm with an automated deep learning CNN feature extraction model to differentiate angiomyolipoma without visible fat (AMLwvf) from ccRCC in CECT images. The model achieved an accuracy of 76.6% for data of 80 samples [65]. Erdim et al. [61] reported that CECT images yielded comparatively superior predictive performance in comparison to unenhanced CT in texture analysis. In the paper, the authors compared results from both unenhanced and contrast-enhanced using different ML algorithms. RF model had the highest accuracy with 88.1% and 90.5% for unenhanced and contrast-enhanced, respectively [61]. For this reason, our paper performed an analysis on the CECT scan.

The present study comprehensively explores the possibility of “virtual biopsy” of renal masses in distinguishing chromophobe renal cell carcinomas from oncocytomas using radiomics and machine learning techniques. The cohorts used were from two different institutions, therefore, they provided some assurance of its external validity, however further research is needed to consolidate this. Our study addressed a specific challenge of distinguishing oncocytic renal masses. Our observations in combination with other reported studies of more common clear cell carcinoma may make it possible to spare patients from more biopsies and move us closer to having a more precise diagnosis. Radiomics based tumor maps, with the ability to capture the patchwork of different types of cancer cells (heterogeneity), may allow clinicians to obtain a more precise tissue sample during biopsies as well. The research in virtual biopsy is growing, and since 2015, publications in this area have doubled [66], as this appeals to two desired end goals in clinical diagnosis of cancers; improved precision and less invasiveness.

There are a few potential limitations in this study that might have had an effect on the results. First, the sample-size was small, primarily because we differentiated rarely occurring tumors so limited data was available. Second, investigation of unenhanced CT images may lead to an analysis of critical internal masses. We did not use unenhanced plain CT to minimize errors in segmentation of the tumors. Due to the rarity of studies in this area, we were unable to validate our results using an independent external data set. Erdim et al. [61] and Lee et al. [67] recommended using deep learning to overcome the problem of false-negative errors in the existing ML classification algorithms, thereby improving the predictive performance in clinical practice. However, this was not implemented in our radiomic engineered-hand-crafted study, as the deep learning model requires a great deal of data to train.

In future research, we propose to explore the use of deep learning models. Likewise, in addition to radiomic signature, other radiomic clusters such as radio-genomics, radio-proteomics and radio-metabolomics need to be studied and compared with the conventional radiomics.

## 5. Summary and Conclusions

This research was to conduct a comparative analysis for the discrimination of ChRCC and RO by the use of ML-based radiomic analysis. The study found that filtered features offer better predictive power compared to original features. Likewise, whole tumor volume data contain more discriminative features compared to 2D slices.

In conclusion, our study established that ML-based radiomic analysis concepts can offer considerable potential for distinguishing ChRCC from RO with a good level of performance. Moreover, filter features, whole tumor volume and prospective study have a competitive edge over original features, 2D and retrospective study, respectively, for the purpose of radiomic analysis. This approach is expected to assist physicians to come up with better diagnoses and improved strategies to tackle the challenges of differentiating RO from ChRCC, and thus help improve oncological precision medicine.

## Figures and Tables

**Figure 1 cancers-14-03609-f001:**
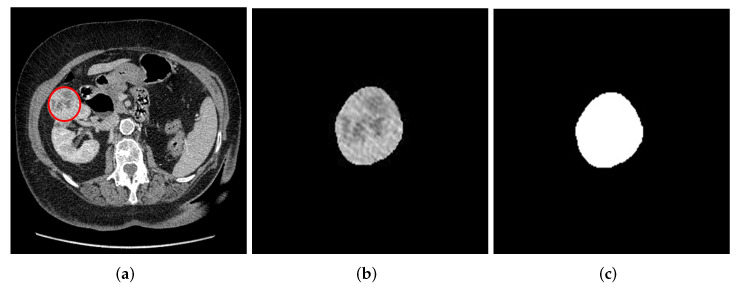
Manual 2D slice image segmentation using image segmentation toolbox in Matlab. (**a**) Original image with ROI. (**b**) Segmented tumor from the kidney. (**c**) Resulting mask from the 2D tumor.

**Figure 2 cancers-14-03609-f002:**
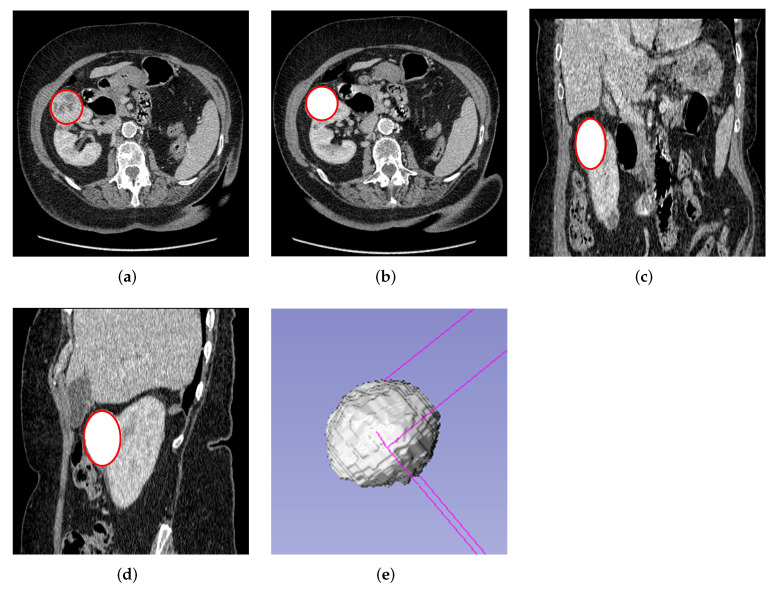
Semi-automatic 3D slice tumor segmentation using the volume editor of the slicer 3D software. (**a**) Original axial plane of the 3D image with VOI. (**b**) Segmented axial tumor plane mask from the kidney. (**c**) Segmented tumor mask of coronal plane. (**d**) Segmented tumor mask of sagittal plane. (**e**) 3D segmentation volume mask.

**Figure 3 cancers-14-03609-f003:**
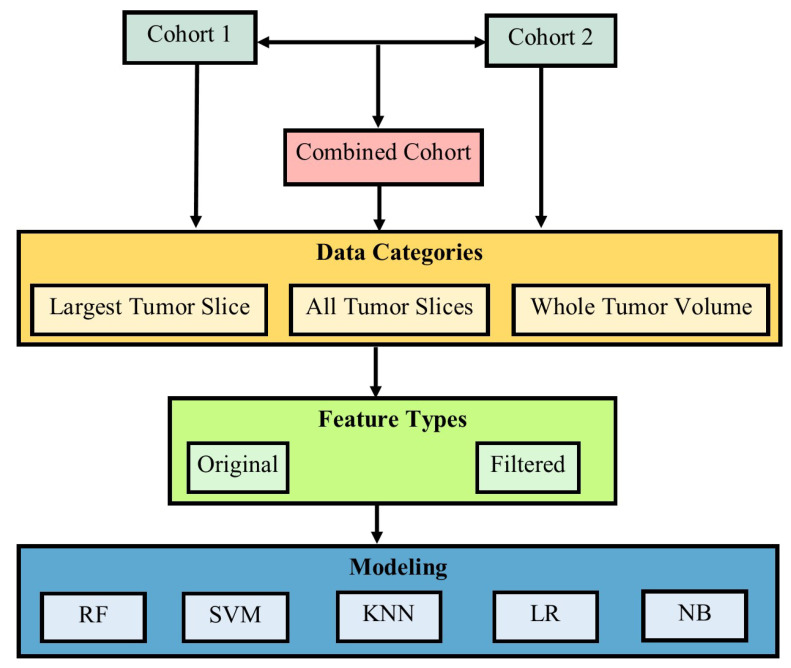
The diagrammatic representation of the strategies involved in the distinction of ChRCC from RO.

**Figure 4 cancers-14-03609-f004:**
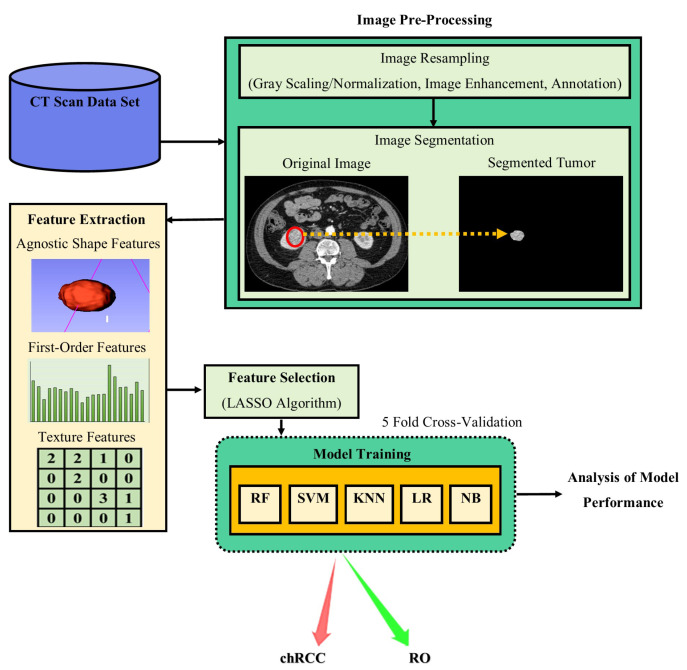
Represents the project methodology process developed by the study.

**Figure 5 cancers-14-03609-f005:**
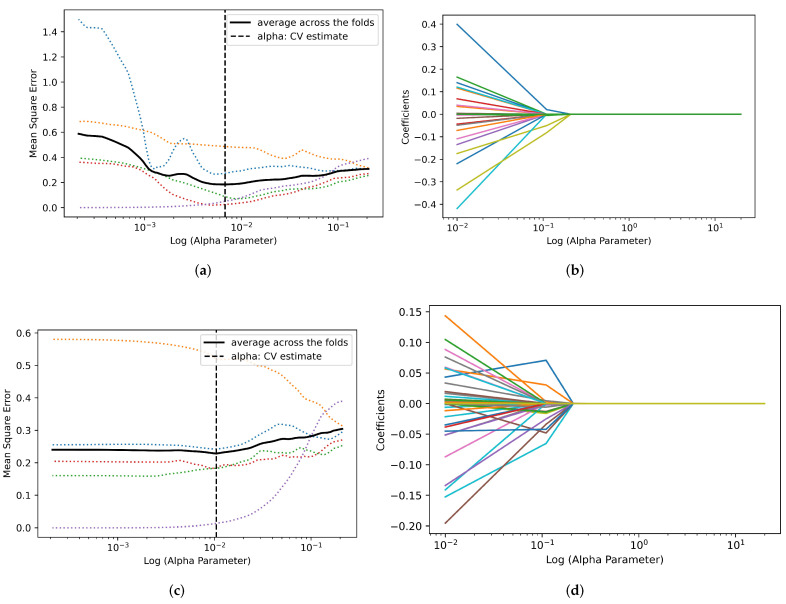
Depiction of LASSO path for Largest Tumor Slice feature selection and best (α) parameter estimation for cohort 1. The black line indicates the mean square error across the five folds. (**a**) Best log (α) estimation for 2D original features. The best (α) estimation parameter was found to be 0.11 presented in vertical black dotted line. (**b**) 2D original feature selection LASSO coefficient path, 4 features were selected. (**c**) Best log (α) estimation parameter for 2D filtered features. The best (α) estimation parameter was found to be 0.11. (**d**) Highlights the 2D filtered feature selection LASSO coefficient path, 14 features were selected.

**Figure 6 cancers-14-03609-f006:**
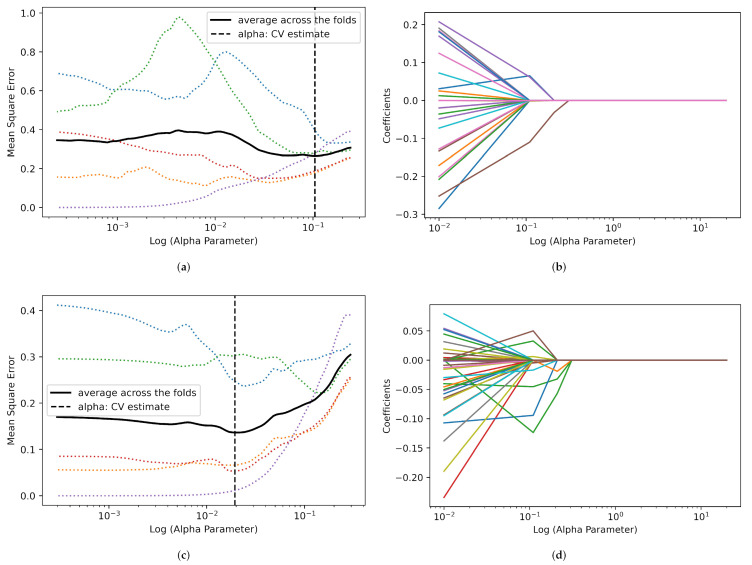
Depiction of LASSO path for 3D Whole Tumor Volume feature selection and best (α) parameter estimation in cohort 1. (**a**) Best log (α) estimation for 3D original features. The best (α) parameter was found to be 0.11. (**b**) 3D original feature selection LASSO path, 4 features were selected. (**c**) Best (α) estimation for 3D filtered features. The (α) parameter was found to be 0.11. (**d**) 3D filtered feature selection LASSO coefficient path, 10 features were selected.

**Table 1 cancers-14-03609-t001:** Description of PyRadiomics feature classes.

PyRadiomic Feature Classes *	
**Class**	**Description**
First-Order	Extracts the voxel-intensities within the region of interest in an image.
GLCM	Describes the quantitative view of texture in an image using image histograms.
GLRLM	Describes the distribution of pixel-intensities in the image run-length features.
GLSZM	Describes the number of related voxels which shares the similar gray-level intensities.
NGTDM	Quantifies the difference between a gray-value and the average gray-value of its neighbors.
GLDM	Describes the gray-level dependencies in an image from its central voxel.
Shape	Describes the dimension shape/size of a volume or region of interest.

^*^ Source: PyRadiomics [47].

**Table 2 cancers-14-03609-t002:** Description of PyRadiomics filter classes.

PyRadiomic Filter Classes *	
**Class**	**Description**
Wavelet	Breaks down an image into frequency component yielding 8 decompositions.
LoG	Enhances areas of rapid intensity changes, i.e., the edges of an image.
Square	Linearly scales the square of image intensity back to their original range.
Square-Root	Linearly scales the square-root of image intensity back to their original range.
Logarithm	Takes the logarithm of absolute-intensities +1 and re-scales them back to their original range.
Exponential	Applies the exponential, in which the filtered intensity is eabsolute intensity).
Gradient	Returns and calculates the magnitude of an image data.
LBP (2D and 3D)	Calculates the local binary pattern in 2D and 3D image using either spherical harmonics or by-slice operation.

^*^ Source: PyRadiomics [47].

**Table 3 cancers-14-03609-t003:** Statistical demographic characteristics of the patients’ data.

	Patients Characteristics				
	**Variable**	**RO**	**ChRCC**	* **p** * **-Value**	**Pearson (r)**
Cohort 1 n=35	Age (Mean ± SD)	69.20 ± 9.27	64.59 ± 10.37	0.22	
	Tumor Size	3.46 ± 1.08	3.52 ± 1.13	0.89	
	Gender			1	
	Male	17 (70.83%)	7 (29.17%)		
	Female	8 (72.73%)	3 (27.27%)		
Cohort 2 n=43	Age (Mean ± SD)	66.56 ± 7.56	53.96 ± 14.49	0.003 *	0.907
	Tumor Size	3.67 ± 1.92	6.09 ± 3.58	0.019 *	0.141
	Gender			0.39	
	Male	7 (50.00%)	7 (50.00%)		
	Female	9 (31.03%)	20 (68.97%)		
Combined n=78	Age (Mean ± SD)	68.17 ± 8.74	56.83 ± 14.30	0 *	0.308
	Tumor Size	3.54 ± 1.47	5.40 ± 3.32	0.002 *	0.107
	Gender			0.11	
	Male	24 (63.16%)	14 (36.84%)		
	Female	17 (42.5%)	23 (57.5%)		

^*^ Statistical significant difference is considered at 0.05 significance level.

**Table 4 cancers-14-03609-t004:** Representation of robust features selected for Largest Tumor Slice.

Largest Tumor Slice			
	**Feature Type**	**Selected Feature Classes**	**n-Features**
Cohort 1	Original	2 GLCM, 1 GLSZM, 1 GLDM	4
	Filtered	7 GLCM, 3 GLSZM, 2 GLDM, 2 First-Order	14
Cohort 2	Original	18 First-Order, 24 GLCM, 32 GLSZM, 14 GLDM, 5 NGTDM	93
	Filtered	12 First-Order, 12 GLCM, 7 GLSZM, 3 GLRLM, 6 NGTDM	40
Combined	Original	1 GLCM, 1 GLRLM, 1 GLSZM	3
	Filtered	1 GLRLM	1

**Table 5 cancers-14-03609-t005:** Representation of robust features selected for All Tumor Slices and Per Patient Prediction.

All Tumor Slices and Per Patient Prediction			
	**Feature Type**	**Feature Classes**	**n-Features**
Cohort 1	Original	7 First-Order, 5 GLCM, 1 GLRLM, 4 GLSZM, 1 GLDM, 2 NGTDM	20
	Filtered	30 First-Order, 27 GLCM, 18 GLSZM, 14 NGTDM, 8 GLDM, 6 GLRLM	103
Cohort 2	Original	8 First-Order, 4 GLCM, 4 GLSZM, 2 GLDM, 1 NGTDM, 1 GLRLM	20
	Filtered	36 First-Order, 24 GLCM, 18 GLSZM, 6 GLRLM, 5 GLDM, 8 NGTDM	97
Combined	Original	5 First-Order, 4 GLCM, 1 GLRLM, 3 GLSZM, 1 NGTDM	14
	Filtered	25 First-Order, 18 GLCM, 14 GLSZM, 4 GLRLM, 2 GLDM, 7 NGTDM	70

**Table 6 cancers-14-03609-t006:** Representation of robust features selected for Whole Tumor Volume.

Whole Tumor Volume			
	**Feature Type**	**Feature Classes**	**n-Features**
Cohort 1	Original	2 First-Order, 1 GLCM, 1 Shape	4
	Filtered	2 First-Order, 2 GLCM, 2 GLDM, 4 GLSZM	10
Cohort 2	Original	4 First-Order, 2 shape, 4 GLCM, 3 GLSZM, 5 GLDM, 2 NGTDM	20
	Filtered	3 First-Order, 1 GLCM, 2 GLDM, 1 NGTDM	7
Combined	Original	1 First-Order, 1 Shape, 1 GLDM	3
	Filtered	2 First-Order, 1 GLRLM, 3 GLDM	6

**Table 7 cancers-14-03609-t007:** Representation of diagnostic performance using largest tumor slice for different models.

Largest Tumor Slice						
	**Feature Type**	**Model**	**Accuracy**	**Sensitivity**	**Specificity**	**AUC**
		RF	0.78 ± 0.115	0.88 ± 0.120	0.68 ± 0.183	0.89 ± 0.087
		SVM	0.66 ± 0.131	0.80 ± 0.157	0.52 ± 0.196	0.76 ± 0.118
	Original	KNN	0.88 ± 0.090	0.96 ± 0.040	0.80 ± 0.157	0.88 ± 0.090
		LR	0.72 ± 0.124	0.76 ± 0.167	0.68 ± 0.183	0.72 ± 0.124
Cohort 1		NB	0.76 ± 0.118	0.76 ± 0.167	0.76 ± 0.167	0.87 ± 0.093
(n=50)		RF	0.92 ± 0.075	0.92 ± 0.080	0.92 ± 0.080	0.95 ± 0.050
		SVM	0.96 ± 0.040	1.00 ± 0.000	0.92 ± 0.080	0.98 ± 0.020
	Filtered	KNN	0.92 ± 0.075	0.96 ± 0.040	0.88 ± 0.120	0.92 ± 0.075
		LR	0.92 ± 0.075	0.96 ± 0.040	0.88 ± 0.120	0.92 ± 0.075
		NB	0.86 ± 0.096	0.84 ± 0.144	0.88 ± 0.120	0.97 ± 0.030
		RF	0.69 ± 0.119	0.63 ± 0.182	0.74 ± 0.166	0.79 ± 0.109
		SVM	0.65 ± 0.126	0.52 ± 0.187	0.78 ± 0.155	0.36 ± 0.128
	Original	KNN	0.70 ± 0.125	0.70 ± 0.176	0.70 ± 0.176	0.73 ± 0.118
		LR	0.63 ± 0.128	0.52 ± 0.187	0.74 ± 0.166	0.63 ± 0.129
Cohort 2		NB	0.56 ± 0.128	0.37 ± 0.183	0.74 ± 0.166	0.60 ± 0.306
(n=54)		RF	0.85 ± 0.097	0.81 ± 0.151	0.89 ± 0.110	0.93 ± 0.068
		SVM	1.00 ± 0.000	1.00 ± 0.000	1.00 ± 0.000	1.00 ± 0.000
	Filtered	KNN	0.89 ± 0.083	0.81 ± 0.151	0.96 ± 0.040	0.89 ± 0.083
		LR	1.00 ± 0.000	1.00 ± 0.000	1.00 ± 0.000	1.00 ± 0.000
		NB	0.81 ± 0.108	0.78 ± 0.155	0.85 ± 0.136	0.85 ± 0.095
		RF	0.61 ± 0.109	0.65 ± 0.148	0.57 ± 0.158	0.65 ± 0.105
		SVM	0.70 ± 0.100	0.70 ± 0.142	0.70 ± 0.142	0.72 ± 0.098
	Original	KNN	0.70 ± 0.100	0.70 ± 0.100	0.70 ± 0.100	0.73 ± 0.097
		LR	0.66 ± 0.106	0.63 ± 0.145	0.70 ± 0.142	0.66 ± 0.104
Combined		NB	0.68 ± 0.098	0.45 ± 0.154	0.90 ± 0.093	0.73 ± 0.097
(n=80)		RF	0.66 ± 0.106	0.65 ± 0.148	0.68 ± 0.140	0.67 ± 0.103
		SVM	0.68 ± 0.098	0.80 ± 0.124	0.55 ± 0.154	0.73 ± 0.097
	Filtered	KNN	0.70 ± 0.078	0.78 ± 0.144	0.63 ± 0.073	0.72 ± 0.107
		LR	0.69 ± 0.099	0.75 ± 0.134	0.62 ± 0.155	0.69 ± 0.101
		NB	0.66 ± 0.106	0.78 ± 0.124	0.55 ± 0.154	0.73 ± 0.097

**Table 8 cancers-14-03609-t008:** Representation of diagnostic performance using all tumor slices for different models.

All Tumor Slices						
	**Feature Type**	**Model**	**Accuracy**	**Sensitivity**	**Specificity**	**AUC**
		RF	0.80 ± 0.023	0.82 ± 0.034	0.78 ± 0.031	0.88 ± 0.018
		SVM	0.67 ± 0.028	0.67 ± 0.038	0.67 ± 0.04	0.73 ± 0.024
	Original	KNN	0.76 ± 0.023	0.88 ± 0.023	0.64 ± 0.038	0.76 ± 0.023
		LR	0.67 ± 0.022	0.66 ± 0.032	0.68 ± 0.033	0.69 ± 0.025
Cohort 1		NB	0.64 ± 0.026	0.74 ± 0.033	0.54 ± 0.036	0.69 ± 0.025
(n=1312)		RF	0.90 ± 0.012	0.89 ± 0.023	0.90 ± 0.025	0.96 ± 0.011
		SVM	0.84 ± 0.016	0.87 ± 0.026	0.80 ± 0.032	0.90 ± 0.016
	Filtered	KNN	0.84 ± 0.023	0.96 ± 0.019	0.72 ± 0.037	0.84 ± 0.020
		LR	0.84 ± 0.016	0.85 ± 0.026	0.82 ± 0.032	0.84 ± 0.020
		NB	0.69 ± 0.023	0.65 ± 0.039	0.72 ± 0.037	0.77 ± 0.023
		RF	0.86 ± 0.014	0.80 ± 0.029	0.91 ± 0.021	0.95 ± 0.009
		SVM	0.76 ± 0.023	0.69 ± 0.031	0.84 ± 0.019	0.83 ± 0.016
	Original	KNN	0.85 ± 0.019	0.79 ± 0.022	0.92 ± 0.016	0.85 ± 0.015
		LR	0.76 ± 0.016	0.71 ± 0.027	0.81 ± 0.019	0.76 ± 0.018
Cohort 2		NB	0.70 ± 0.014	0.51 ± 0.027	0.88 ± 0.021	0.75 ± 0.018
(n=2152)		RF	0.91 ± 0.011	0.88 ± 0.015	0.94 ± 0.015	0.98 ± 0.006
		SVM	0.85 ± 0.012	0.82 ± 0.020	0.88 ± 0.017	0.92 ± 0.011
	Filtered	KNN	0.90 ± 0.010	0.84 ± 0.025	0.95 ± 0.014	0.85 ± 0.015
		LR	0.85 ± 0.018	0.83 ± 0.023	0.87 ± 0.024	0.85 ± 0.015
		NB	0.74 ± 0.023	0.58 ± 0.032	0.91 ± 0.014	0.82 ± 0.016
		RF	0.79 ± 0.019	0.75 ± 0.019	0.84 ± 0.020	0.88 ± 0.012
		SVM	0.72 ± 0.016	0.74 ± 0.027	0.69 ± 0.028	0.80 ± 0.015
	Original	KNN	0.76 ± 0.023	0.74 ± 0.021	0.79 ± 0.022	0.84 ± 0.014
		LR	0.71 ± 0.020	0.72 ± 0.024	0.71 ± 0.021	0.71 ± 0.017
Combined		NB	0.72 ± 0.013	0.76 ± 0.023	0.67 ± 0.026	0.77 ± 0.016
(n=2774)		RF	0.84 ± 0.014	0.79 ± 0.022	0.89 ± 0.016	0.92 ± 0.010
		SVM	0.78 ± 0.018	0.74 ± 0.022	0.83 ± 0.017	0.85 ± 0.013
	Filtered	KNN	0.81 ± 0.019	0.76 ± 0.022	0.87 ± 0.018	0.81 ± 0.015
		LR	0.78 ± 0.018	0.75 ± 0.025	0.81 ± 0.012	0.78 ± 0.015
		NB	0.72 ± 0.017	0.79 ± 0.018	0.65 ± 0.028	0.81 ± 0.015

**Table 9 cancers-14-03609-t009:** Representation of diagnostic performance using per patient prediction from all tumor slices using majority voting technique for different models.

Per Patient Prediction						
	**Feature Type**	**Model**	**Accuracy**	**Sensitivity**	**Specificity**	**AUC**
		RF	1.00 ± 0.000	1.00 ± 0.000	1.00 ± 0.000	1.00 ± 0.000
		SVM	0.71 ± 0.154	0.60 ± 0.304	0.76 ± 0.167	0.68 ± 0.155
	Original	KNN	0.91 ± 0.090	1.00 ± 0.000	0.88 ± 0.120	0.94 ± 0.060
		LR	0.66 ± 0.154	0.60 ± 0.304	0.68 ± 0.183	0.64 ± 0.159
Cohort 1		NB	0.66 ± 0.154	0.90 ± 0.100	0.56 ± 0.195	0.73 ± 0.147
(n=35)		RF	0.97 ± 0.030	1.00 ± 0.000	0.96 ± 0.040	0.98 ± 0.020
		SVM	0.97 ± 0.030	1.00 ± 0.000	0.96 ± 0.040	0.98 ± 0.020
	Filtered	KNN	0.91 ± 0.090	1.00 ± 0.000	0.88 ± 0.120	0.94 ± 0.060
		LR	0.63 ± 0.159	0.60 ± 0.304	0.64 ± 0.188	0.62 ± 0.161
		NB	0.74 ± 0.148	0.70 ± 0.284	0.76 ± 0.167	0.73 ± 0.147
		RF	0.67 ± 0.139	0.59 ± 0.188	0.80 ± 0.200	0.70 ± 0.139
		SVM	0.60 ± 0.144	0.48 ± 0.190	0.80 ± 0.200	0.64 ± 0.145
	Original	KNN	0.74 ± 0.188	0.70 ± 0.176	1.00 ± 0.000	0.85 ± 0.108
		LR	0.60 ± 0.144	0.48 ± 0.190	0.80 ± 0.200	0.64 ± 0.145
Cohort 2		NB	0.55 ± 0.148	0.30 ± 0.169	1.00 ± 0.000	0.65 ± 0.144
(n=42)		RF	0.76 ± 0.131	0.70 ± 0.176	0.87 ± 0.130	0.79 ± 0.123
		SVM	0.71 ± 0.141	0.63 ± 0.182	0.87 ± 0.130	0.75 ± 0.131
	Filtered	KNN	0.86 ± 0.103	0.78 ± 0.155	1.00 ± 0.000	0.89 ± 0.095
		LR	0.67 ± 0.139	0.63 ± 0.182	0.73 ± 0.227	0.68 ± 0.141
		NB	0.57 ± 0.151	0.37 ± 0.183	0.93 ± 0.070	0.65 ± 0.144
		RF	0.73 ± 0.097	0.59 ± 0.163	0.85 ± 0.111	0.72 ± 0.100
		SVM	0.68 ± 0.100	0.73 ± 0.143	0.63 ± 0.145	0.68 ± 0.104
	Original	KNN	0.69 ± 0.102	0.59 ± 0.163	0.78 ± 0.124	0.70 ± 0.102
		LR	0.64 ± 0.104	0.59 ± 0.163	0.68 ± 0.140	0.63 ± 0.108
Combined		NB	0.68 ± 0.100	0.73 ± 0.143	0.63 ± 0.145	0.65 ± 0.107
(n=77)		RF	0.75 ± 0.100	0.65 ± 0.152	0.85 ± 0.111	0.76 ± 0.095
		SVM	0.69 ± 0.102	0.60 ± 0.153	0.78 ± 0.124	0.68 ± 0.104
	Filtered	KNN	0.82 ± 0.084	0.76 ± 0.135	0.88 ± 0.097	0.82 ± 0.086
		LR	0.70 ± 0.104	0.59 ± 0.163	0.80 ± 0.124	0.70 ± 0.206
		NB	0.68 ± 0.100	0.73 ± 0.143	0.63 ± 0.145	0.65 ± 0.107

**Table 10 cancers-14-03609-t010:** Representation of diagnostic performance using whole tumor volume for different models.

Whole Tumor Volume						
	**Feature Type**	**Model**	**Accuracy**	**Sensitivity**	**Specificity**	**AUC**
		RF	0.84 ± 0.102	0.92 ± 0.080	0.76 ± 0.167	0.93 ± 0.070
		SVM	0.88 ± 0.090	0.96 ± 0.040	0.80 ± 0.157	0.89 ± 0.087
	Original	KNN	0.88 ± 0.090	0.84 ± 0.144	0.92 ± 0.080	0.88 ± 0.090
		LR	0.80 ± 0.111	0.76 ± 0.167	0.84 ± 0.144	0.80 ± 0.111
Cohort 1		NB	0.78 ± 0.115	0.72 ± 0.760	0.84 ± 0.144	0.85 ± 0.099
(n=50)		RF	0.96 ± 0.040	0.96 ± 0.040	0.96 ± 0.040	1.00 ± 0.000
		SVM	0.86 ± 0.096	0.88 ± 0.120	0.84 ± 0.144	0.98 ± 0.020
	Filtered	KNN	0.94 ± 0.060	1.00 ± 0.000	0.88 ± 0.120	0.94 ± 0.060
		LR	0.92 ± 0.075	1.00 ± 0.000	0.84 ± 0.144	0.92 ± 0.075
		NB	0.92 ± 0.075	0.92 ± 0.080	0.92 ± 0.080	0.98 ± 0.020
		RF	0.76 ± 0.113	0.70 ± 0.176	0.81 ± 0.151	0.88 ± 0.087
		SVM	0.80 ± 0.104	0.78 ± 0.155	0.81 ± 0.151	0.88 ± 0.087
	Original	KNN	0.72 ± 0.122	0.74 ± 0.166	0.70 ± 0.176	0.76 ± 0.114
		LR	0.83 ± 0.103	0.78 ± 0.155	0.89 ± 0.110	0.84 ± 0.098
Cohort 2		NB	0.67 ± 0.122	0.59 ± 0.188	0.74 ± 0.166	0.77 ± 0.112
(n=54)		RF	0.83 ± 0.103	0.78 ± 0.155	0.89 ± 0.110	0.93 ± 0.068
		SVM	0.87 ± 0.090	0.82 ± 0.141	0.93 ± 0.070	0.94 ± 0.060
	Filtered	KNN	0.85 ± 0.097	0.85 ± 0.136	0.85 ± 0.136	0.89 ± 0.083
		LR	0.81 ± 0.108	0.78 ± 0.155	0.85 ± 0.136	0.82 ± 0.102
		NB	0.78 ± 0.109	0.81 ± 0.151	0.74 ± 0.166	0.90 ± 0.080
		RF	0.66 ± 0.101	0.63 ± 0.152	0.68 ± 0.145	0.70 ± 0.099
		SVM	0.68 ± 0.104	0.59 ± 0.146	0.78 ± 0.127	0.72 ± 0.097
	Original	KNN	0.70 ± 0.095	0.51 ± 0.155	0.88 ± 0.098	0.67 ± 0.102
		LR	0.70 ± 0.095	0.63 ± 0.152	0.76 ± 0.128	0.70 ± 0.099
Combined		NB	0.71 ± 0.096	0.46 ± 0.156	0.95 ± 0.050	0.74 ± 0.095
(n=82)		RF	0.71 ± 0.096	0.66 ± 0.144	0.76 ± 0.128	0.80 ± 0.087
		SVM	0.77 ± 0.090	0.76 ± 0.128	0.78 ± 0.127	0.84 ± 0.079
	Filtered	KNN	0.80 ± 0.091	0.78 ± 0.127	0.83 ± 0.114	0.87 ± 0.073
		LR	0.77 ± 0.090	0.76 ± 0.128	0.78 ± 0.127	0.77 ± 0.091
		NB	0.78 ± 0.090	0.71 ± 0.137	0.85 ± 0.112	0.85 ± 0.077

**Table 11 cancers-14-03609-t011:** Summary of best diagnostic performance. The table displays the best AUCs from the different categories and cohorts.

	Largest Tumor Slice			
**Cohorts**	**Feature Type**	**Model**	**Feature Selected**	**AUC**
1	Filter	SVM	14	0.98 ± 0.02
2	Filter	SVM	40	1.00 ± 0.000
Combined	Original	KNN	3	0.73 ± 0.097
	**All Tumor Slices**			
1	Filter	RF	103	0.96 ± 0.011
2	Filter	RF	97	0.98 ± 0.006
Combined	Filter	RF	70	0.92 ± 0.010
	**Per Patient Prediction from All Tumor Slices**			
1	Original	RF	20	1.00 ± 0.000
2	Filter	KNN	97	0.89 ± 0.095
Combined	Filter	KNN	70	0.82 ± 0.086
	**Whole Tumor Volume**			
1	Filter	RF	10	1.00 ± 0.000
2	Filter	SVM	7	0.94 ± 0.060
Combined	Filter	KNN	6	0.87 ± 0.073

## Data Availability

The data provided in the prospective cohort 1 study are available on request from the corresponding author. For cohort 2, the data is readily available from the Kits GitHub page and Cancer Imaging Archive (CIA) [34,35]. The codes used to reproduce the results can be found on Github upon request at this link https://github.com/abeer2005/Classification-of-chromophobe-and-oncocytoma-using-Radiomic-Feature-Analysis-and-Machine-Learning (accessed on 29 January 2022).

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
