# Peer review of "Comparative Analysis for the Distinction of Chromophobe Renal Cell Carcinoma from Renal Oncocytoma in Computed Tomography Imaging Using Machine Learning Radiomics Analysis"

_cancers, 2022, doi:10.3390/cancers14153609_

Round 1

Reviewer 1 Report

This manuscript analyses the potential of Machine Learning to discriminate chromophobe renal cell carcinoma from renal oncocytoma in Computed Tomography Imaging using Radiomics

The study is well conducted to analyze the basic priniciples of radiomics in this cohort. However, some limitations apply mainly concerning the study design

Major:

Study Design:

-       Why do the authors try to distinguish chrRRC and oncocytoma and not renal tumors in general. The current concept does not change management of a renal mass from a clinical perspective. The current challenge of radiomics in renal masses is to diagnose correctly in unfiltered cohorts. Please comment

Minor:

Introduction:

-       Does RCC really have the highest mortality rate? Is urothelial carcinoma not higher? 

-       The authors change between the terms “renall cell carcinoma” and “kidney cancer”. Please be precise.

Methods

-       Segmentation: Was the tumor margin included or excluded from the ROI.

Discussion

-       The clinical relevance of the findings is not clearly outlined and mainly technical aspects are discussed. Some of the aspects are already covered by contemporary recommendations of international radiomics classifications.

Reviewer 2 Report

Your paper is fairly well written and shows a new ML-based radiomic analysis for the differentiation of renal oncocytoma and chromophobe carcinoma. The followings need a clarification to improve your paper.

1. Oncocytoma often shows iso- or hyperdensity on noncontrast CT. How does this compare with chromophobe renal carcinoma in your study?

2. Central stellate scar has been noted in 33% of renal oncocytoma. How does this observe in chromophobe renal carcinoma in your study?

3. Line 32: Kidney------>kidney or renal

4. Line 327: Per Patient Prediction------>per patient prediction

5. Line 329: Largest Tumor Slice------->largest tumor slice

6 Lines 330 and 332: Whole Tumor Volume-------->whole tumor volume

Round 2

Reviewer 1 Report

The authors replied to the comments sufficiently.